# Construction of the experimental rat model of gestational diabetes

**Fan Chen**[1], **Li Ge**[1]*, **Xinyong Jiang**[1], **Yuting Lai**[1], **Pingping Huang**[1], **Jinghe Hua**[1], **Yuzheng Lin**[1], **Yan Lin**[1], **Xiumin Jiang**[2]

**1** School of Nursing, Fujian University of Traditional Chinese Medicine, Fuzhou, China, **2** Fujian Maternity and Child Health Hospital Affiliated to Fujian Medical University, Fuzhou, China

* 2000005@fjtcm.edu.cn

**Data Availability Statement:** All supporting Information files are available from the figshare database (accession website link: https://doi.org/10.6084/m9.figshare.19802617.v1).

## Abstract

### Objective

Numerous methods for modeling gestational diabetes mellitus (GDM) in rats exist. However, their repeatability and stability are unclear. This study aimed to compare the effects of high-fat and high-sugar (HFHS) diet, HFHS diet combined with streptozotocin (STZ) administration, and HFHS diet combined with movement restriction (MR) modeling methods on rat models to confirm the best method for constructing a rat model of GDM.

### Method

Forty female Sprague-Dawley rats were randomly divided into four groups (n = 10): the normal control (NC), HFHS, HFHS+STZ, and HFHS+MR groups. The rats in the NC group were fed with a standard diet, and those in the remaining groups were fed with a HFHS diet. The rats in the HFHS+STZ group received 25 mg/kg STZ on their first day of pregnancy, and those in the HFHS+MR group were subjected to MR during pregnancy. Bodyweight, food intake, water intake, fasting blood glucose (FBG), fasting insulin (FINS), homeostasis model assessment of insulin resistance (HOMA-IR), homeostasis model assessment of insulin sensitivity (HOMA-IS), homeostasis model assessment of β-cell function, pancreatic and placental morphology, and the expression levels of glucose transporter 1 (GLUT1) and glucose transporter 3 (GLUT3) in placentas were then quantified. Moreover, iTRAQ was used to identify placental proteomics.

### Results

During pregnancy, the rats in the HFHS+STZ group showed FBG levels that were kept stable in a state of moderate hyperglycemia; the typical GDM symptoms of polydipsia, polyphagia, polyuria, and increased body weight; and the modeling rate of 87.5%. On the first and 19th days of pregnancy, the rats in the HFHS group showed higher FBG than that of the NC group, increasing body weight and food intake and the modeling rate of 50%. On the 19th day of pregnancy, the FBG of the rats in the HFHS+MR group was higher than that of the rats in the NC group, and the modeling rate of 42.9%. Comparison with the NC group revealed that the three modeling groups exhibited increased FINS and HOMA-IR,

**Funding:** Funding was provided by the Fujian University of Traditional Chinese Medicine (funding number: X2019041-discipline). The funders had no role in study design, data collection and analysis, decision to publish, or preparation of the manuscript. We appreciate your consideration.

**Competing interests:** The authors have declared that no competing interests exist.

decreased HOMA-IS, and different degrees of pathological changes in pancreases and placentas. Among the groups, the HFHS+STZ group displayed the greatest changes with significant reductions in the numbers of pancreatic and placental cells and appeared cavitation. The expression levels of GLUT1 and GLUT3 in the placentas of the HFHS+STZ and HFHS+MR groups were higher than those in the placentas of the NC and HFHS groups. The above results indicated that the rats in the HFHS+STZ group showed the best performance in terms of modeling indicators. After the changes in placental proteomics in the HFHS+STZ group were compared with those in the NC group, we found that in the HFHS+STZ group, five proteins were up-regulated and 18 were down-regulated; these proteins were enriched in estrogen signaling pathways.

## Conclusion

HFHS combined with the intraperitoneal injection of 25 mg/kg STZ was the best modeling method for the nonspontaneous model of experimentally induced GDM, and its modeling rate was high. The pathological characteristics of the constructed GDM rat model were similar to those of human patients with GDM. Moreover, the model was stable and reliable. The modeling method can provide a basis for constructing a GDM rat model for subsequent research on the prevention and treatment of GDM.

## Introduction

The latest (2019) estimate of the International Diabetes Federation states that gestational diabetes mellitus (GDM) affects approximately 13.2% of pregnancies worldwide [1]. GDM can pose short-term or long-term health risks to mothers and fetuses, including stillbirth, type 2 diabetes, and obesity [2]. Therefore, it is urgent to find better methods and strategies to prevent and treat GDM.

An ideal GDM animal model is an essential tool for studying the pathogenesis, causes, and development of GDM and discovering and optimizing GDM prevention and treatment methods. Given the extensive application of rat models in diabetes and the shorter pregnancy, maturity, and growth periods of rats than those of other experimental animals, such as dogs, sheep, pigs, and primates, rats are widely used to construct GDM models. Current GDM rat models include spontaneous or genetically derived models (BBDP/BBDR rats, Zucker fatty rats, LepR[db] rats and Goto-Kakizaki rats) and experiment-induced nonspontaneous models [3]. The latter is commonly used in medical research due to its relatively low cost, short experimental period, and facile induction and disease control [4].

Numerous known methods exist for constructing experimentally induced nonspontaneous GDM rat models, including diet induction, chemical drug induction, and chemical drug induction combined with diet induction [5]. The high-fat and high-sugar (HFHS) diet is thus widely used because it can not only simulate the unhealthy dietary mode of humans [6] and induce the apparent symptoms of hyperglycemia and hyperinsulinemia in rats but also cause obesity, reduce the sensitivity of insulin target organs to insulin, thereby inducing insulin resistance (IR) and other pathological changes similar to human GDM [7, 8]. Streptozotocin (STZ) is currently the most widely utilized chemical inducer of diabetes animal models [9]. As a glucose analog, STZ can be absorbed by pancreatic β cells via glucose transporter 2 and inhibit DNA synthesis by inducing DNA division and methylation, thus leading to pancreatic β cell

apoptosis and causing hyperglycemia symptoms [10]. Although the administration of low doses of STZ to rats on a regular diet can slightly impair insulin secretion [11], it does not significantly change blood sugar and IR levels [12, 13]. Administering low doses of STZ to rats fed with a HFHS diet slightly impairs insulin secretion and causes a pathogenic process similar to the pathogenesis of human GDM [14, 15]. Therefore, the HFHS diet combined with a single intraperitoneal injection of low-dose STZ has become a common choice for constructing GDM rat models in recent years. In addition to changes in dietary structure, reduced physical activity and sedentary behavior during pregnancy are the main reasons for the morbidity of GDM [16]. The provision of rooms that can satisfy movement, rest, and regular posture adjustment to rats not only mimics the effects of long durations of sitting in humans but also results in the conspicuous symptoms of increased body fat and postprandial hyperglycemia [17]. Therefore, the modeling method that simulates dietary changes and sedentary behavior is in line with the physiological and pathological characteristics of pregnant women with GDM. However, no research on applying the HFHS diet combined with movement restriction (MR) to construct GDM rat models exists.

Moreover, we have found significant differences in glycolipid ratios, compositions, feeding time, and chemical drug injection doses among previous GDM rat modeling methods [8, 12]. The lack of the comprehensive and in-depth evaluation of GDM rat models from the perspectives of biochemical blood metabolism, histomorphology, and molecular biology makes the reliability, stability, and repeatability of GDM rat models uncertain. Therefore, comprehensively evaluating and determining a highly appropriate, reliable, and stable GDM rat model is of great importance.

This study aimed to compare three methods for constructing GDM rat models, namely, the HFHS diet, the HFHS diet combined with the intraperitoneal injection of 25 mg/kg STZ, and the HFHS diet combined with MR, in order to confirm the best method for constructing a rat model of GDM which is suitably, reliable, stable and is similar to human GDM. Such a model will provide a reliable tool for future experimental research on GDM.

## Materials and methods

### Ethics statement

The study was performed in accordance with the guidelines for the ethical review of laboratory animal welfare in the People's Republic of China National Standard (GB/T 35892–2018) and approved by the Institutional Animal Care and Use Committee of Fujian University of Traditional Chinese Medicine (Approval NO.FJTCM IACUC 2019051). In order to alleviate rats' suffering as far as possible, rats were sacrificed under deep anesthesia by intraperitoneal injection with 20% urethane (1 g/kg body weight) by rapid exsanguination through cutting off the abdominal aorta after collecting sufficient abdominal aortic blood.

### Animals and diet components

Forty female and 20 male specific pathogen-free (SPF) Sprague–Dawley rats at ten weeks of age were obtained from the Experimental Animal Center of Zhejiang Academy of Medical Sciences (Production license number: SCXK [Zhe]2019–0002). The male and female rats were separated and housed five per cage in an SPF-grade laboratory maintained at 23°C– 27°C and 55%– 65% humidity under a 12-hour light-dark cycle with ad libitum access to standard rodent diet and water.

The HFHS diet was composed of 66.5% basic feed, 20.0% sucrose, 10.0% cooked lard, 2.5% cholesterol, and 1.0% cholat, and the total energy obtained from the HFHS diet was 4.43 kcal/g in which 34.42%, 12.65%, and 52.93% were derived from fat, protein, and carbohydrates,

respectively [8]. The total energy of the standard diet was 3.42 kcal/g in which 12.11%, 22.47%, and 65.42% were derived from fat, protein, and carbohydrates, respectively (GB 14924 1022). Food was provided by Minhou County Wushi Experimental Animal Trade Co., Ltd.

## Construction of GDM models

After one week of acclimatization, the female rats were randomized into four groups (n = 10/ group) using a random number table method: the normal control (NC), HFHS, HFHS+STZ, and HFHS+MR groups. Before mating, the rats in the NC group were fed with the standard diet for six weeks, and those in the three other groups were fed with the HFHS diet for six weeks. During this period, hyperglycemic rats with blood glucose levels higher than 6.7 mmol/ L (120 mg/dL) were excluded [18]. After six weeks, the female and male rats were mated at a 2:1 ratio, and the first day of pregnancy was confirmed by examining vaginal smears. Rats were excluded from the subsequent study if they were not pregnant one week after mating. During the mating and pregnancy periods, the rats in the NC group were continuously fed with the standard diet, and those in the three other groups were continuously fed with the HFHS diet. Moreover, the rats in the HFHS+STZ group were fasted for 8 h from the discovery of pregnancy and then given a single intraperitoneal injection of 25 mg/kg STZ (Macklin, diluted with 0.1 M sodium citrate buffer, pH 4.5). The activity space of the rats in the HFHS +MR group was reduced 2/3 by placing a baffle made of acrylic plastic in the site of 1/3 cage, where was L: 395 cm × W: 200 cm × H: 200 cm and did not affect the rats normally taking food and water. Rats with tail vein fasting blood glucose (FBG) levels greater than 6.7 mmol/L on the fourth day of pregnancy were used as the GDM models.

## FBG and modeling rate

After 8 h of fasting on the first, fourth, seventh, 14th, and 19th days of pregnancy, a blood glucose meter was used to measure the tail vein FBG of the rats. On the fourth day of pregnancy (after STZ injection and 72 h of MR), the modeling rate of each group was calculated in accordance with the modeling standard of FBG $\geq$ 6.7 mmol/L as follows: modeling rate = the number of model rats/number of pregnant rats × 100%.

## General condition

During the experimental period, hair luster, activity, urine output, food intake and water intake were monitored daily, and body weight was recorded weekly.

## FINS, HOMA-IR, HOMA-IS, and HOMA-β

On the 19th day of pregnancy, all rats were fasted for 12 h and then injected with 1 g/kg 20% urethane solution (Sigma) into their abdominal cavities. It showed that the rats were in deep anesthesia when the rats did not respond after their toes of the hind limbs were squeezed, and then the follow-up experiments were started. Blood samples were collected from the abdominal aorta. Serum was separated via centrifugation at 3000 rpm for 10 min, aliquoted into EP tubes, and stored in a refrigerator at −80˚C. Serum fasting insulin (FINS) was measured using a rat insulin ELISA kit (Jiangsu Feiya Biotechnology Co., Ltd., China). Homeostasis model assessment of insulin resistance (HOMA-IR), homeostasis model assessment of insulin sensitivity (HOMA-IS), homeostasis model assessment of β-cell function (HOMA-β) were calculated by using the following mathematical formulae: HOMA-IR = FBG × FINS/22.5; HOMA-IS = 22.5/(FBG × FINS); HOMA-β = 20 × FINS/(FBG − 3.5) [19].

## Structure and morphology of the pancreas and placenta

After the rats were sacrificed via bloodletting under deep anesthesia, the pancreatic and placental tissues were immediately removed, fixed in 4% paraformaldehyde fixative (Shanghai Beyotime Biotechnology Co., Ltd., China) for 24 h, and embedded in paraffin. The paraffin-embedded tissues were serially sectioned at 5 μm and stained with hematoxylin and eosin (HE). The HE-stained sections were observed under 100× magnification.

## Protein expression levels of GLUT1 and GLUT3

After the rats were sacrificed, the placental tissues were collected via laparotomy, quickly frozen with liquid nitrogen, and stored in a refrigerator at −80˚C. Placental samples were lysed in RIPA buffer containing protease inhibitor cocktails. The total protein was quantified by using a BCA kit (Beyotime Biotechnology, Shanghai, China). The placental protein lysates (100 mg) were separated by using 10% SDS-PAGE (75 V constant, 128 min) and electrotransferred onto PVDF membranes (300 mA constant, 74 min). After blocking with 5% skimmed milk in PBS-Tween, the membranes were probed with anti-glucose transporter 1 (GLUT1) or anti-glucose transporter 1 (GLUT3) primary antibody (1:500, Boeter) at 4˚C overnight and conjugated with secondary antibodies at room temperature for 45 min. Antibodies against GLUT1 and GLUT3 were obtained from BOSTER Biological Technology Co., Ltd. The membranes were visualized with an ECL chemiluminescence detection kit (Jiangsu KeyGEN Biotechnology Co., Ltd., China). β-Actin expression was used as the loading control for each sample. Optical densities were scanned and analyzed by using Image J software (Bandscan 5.0).

## Protein extraction, trypsin digestion, and iTRAQ labeling

A total of 100 mg of placental tissue was thoroughly ground into cell powder with liquid nitrogen and then mixed with four volumes of lysis buffer (1% Triton X-100, 1% Protease Inhibitor Cocktail) for ultrasonic lysis. The sample was centrifuged at 12 000 rpm at 4˚C for 10 min. Its supernatant was collected, and the protein concentration was determined with a BCA kit. Then, 20% trichloroacetic acid was slowly added to the same amount of protein sample. The sample was mixed through vortexing, precipitated at 4˚C for 2 h, and centrifuged at 4500 rpm for 5 min. The supernatant was discarded, and the precipitant was washed with precooled acetone for 2–3 times. The precipitant was dried, added with TEAB at the final concentration of 200 mM, dispersed ultrasonically, and added with pancreatin at 37˚C to prepare a protein suspension culture (1:50). Dithiothreitol was added to a final concentration of 5 mM, and the sample was deacidized at 56˚C for 30 min. Then, iodoacetamide was added at the final concentration of 11 mM, and the sample was incubated for 15 min at room temperature in the dark. After trypsin digestion, peptides were desalted with Strata X C18 SPE column (Phenomenex) and vacuum-dried. The peptides were dissolved with 0.5 M TEAB and labeled according to the TMT Sixplex™ Labeling Kit (Thermo Fisher) instructions. The three NC group samples were labeled as 113, 114, and 115, and the three HFHS+STZ group samples were labeled as 116, 117, and 118 with the iTRAQ reagents.

## HPLC fractionation and liquid chromatography–mass spectrometry

An Agilent 300 Extend C18 column (5 μm particle size, 4.6 mm inner diameter, 250 mm length) was used to fractionate peptides via high-pH reversed-phase HPLC with a gradient of 8%–32% acetonitrile (pH 9.0). Sixty fractions were separated within 60 min. The peptides were combined into 14 fractions and freeze-dried by vacuum centrifuging. A total of 10 μL of the sample was taken for liquid chromatography-mass spectrometry analysis. The liquid-phase

system was EASY-nLC 1000 (America, Thermo Fisher Scientific, America, #LC120). Mobile phase A was an aqueous solution containing 0.1% formic acid and 2% acetonitrile, and mobile phase B was an aqueous solution containing 0.1% formic acid and 90% acetonitrile. The gradient was as follows: Phase B was increased linearly from 8% to 22% in 20 min; from 22% to 35% in 13 min; climbed to 80% in 4 min; and held at 80% for the last 3 min, all at a constant flow rate of 600 nL/min. After the liquid phase was separated, the peptides were injected into the NSI ion source for ionization and then into QExactive TM plus for mass spectrometer analysis. The ion source voltage was 2.2 kV. The MS system was a high-resolution Orbitrap system. Data were collected by using a data-dependent scanning program.

## Proteomics database search

The Maxquant search engine (v.1.5.2.8) and the following search parameters were used to search the Rattus_norvegicus_10116 database, and the contaminating protein sequence in Uniprot: iTRAQ 8 plex (peptide tag) was used as the quantitative method. Trypsin/P was specified as a cleavage enzyme allowing up to 2 missing cleavages. The mass tolerance for precursor ions was set as 10 ppm in First search and 5 ppm in Main search, and the mass tolerance for fragment ion was set as 0.02 Da. Carbamidomethyl on Cys was set for fixed modification, and ('Acetyl [Protein N-term]', 'Oxidation [M]', 'Deamidation [NQ]') was set for variable modification. Only proteins with unused values exceeding 1.3 were considered for further analysis. The data were collected on the basis of protein identification with the false positive rate $\leq 1\%$ and peptide–spectral match confidence interval.

## Bioinformatics analysis

Gene Ontology (GO) annotation was performed to understand the biological functions of differentially expressed placental proteins. The annotations included biological processes (BP), cell components (CC), and molecular functions (MF) (http://www.ebi.ac.uk/GOA/). The Perl module tool (https://metacpan.org/pod/Text:Text::NSP::Measures::2D::Fisher) was applied for the enrichment analysis of the function of the differentially expressed proteins. The Kyoto Encyclopedia of Genes and Genomes (KEGG) (http://www.genome.jp/kegg/) database was utilized for the enrichment analysis of the signaling pathways of the differentially expressed proteins.

## Statistical analysis

IBM SPSS Statistics 26.0 software was used to analyze the data, and Graph Pad Prism 8 was applied to draw graphs. Data with normal distribution and uniform variances were expressed as mean ± standard deviation. Multiple groups were compared via one-way analysis of variance. Data that were not normally distributed or had uneven variance were expressed in terms of median and interquartile range (M[Q25, Q75]). Kruskal–Wallis was used for comparison among multiple groups. $P < 0.05$ was considered as a significant difference.

## Results

### FBG changes and modeling rate

A total of 40 rats were included in this study, among which 29 were successfully impregnated, and none of the rats died. Among the pregnant rats, six were in the NC group, eight were in the HFHS group, eight were in the HFHS+STZ group, and seven were in the HFHS+MR group. During pregnancy, the FBG of the rats in the three modeling groups increased relative to that of the rats in the NC group, as followed: The FBG of the rats in the HFHS group increased significantly on the first and 19th days of pregnancy; that of the rats in the HFHS

+STZ group increased significantly on the fourth, seventh, 14th, and 19th days of pregnancy; and that of the rats in the HFHS+MR group increased significantly on the 19th day of pregnancy (Fig 1). The rat modeling rates of the HFHS, HFHS+STZ and HFHS+MR groups were 50%, 87.5%, and 42.9%, respectively. These results showed that the three investigated modeling methods could be used to construct GDM rat models. The HFHS+STZ group maintained a relatively stable FBG and had a higher model formation among the groups.

## General condition

Before pregnancy, there was no significant difference in body weight and water intake among the four groups; compared with the rats in the NC group, the food intake of other three groups was significantly increased, but there was no significant difference in the three groups. Compared with the rats in the NC group, the rats in the HFHS group showed increased body weight on the first and 19th days of pregnancy, increased food intake during pregnancy, and no significant changes in water intake. The rats in the HFHS+STZ group displayed ungroomed fur, polyuria, inactivity, increased body weight, food intake and water intake during pregnancy. The rats in the HFHS+MR group exhibited ungroomed fur, listlessness, and inactivity during pregnancy, and decreased water intake on the seventh, 14th, and 19th days of pregnancy, and no significant difference in body weight and food intake. Compared the three modeling groups, the body weight of the rats in the HFHS group was higher than that of the rats in the HFHS+MR group on the 19th day of pregnancy (Fig 2A). On the seventh, 14th, and 19th day of pregnancy, the rats in the HFHS+STZ group had higher food intake than those in the HFHS+MR group (Fig 2B), and had higher water intake than those in the HFHS and HFHS +MR groups; at the same point in time, the rats in the HFHS group had higher water intake than those in the HFHS+MR group (Fig 2C). These results indicated that although the three modeling methods could induce changes in the general condition of the pregnant rats, the pregnant rats under induction by the HFHS diet combined with STZ injection exhibited the typical GDM signs of polydipsia, polyphagia, polyuria, bodyweight gain, ungroomed fur, listlessness, and inactivity. The HFHS-induced pregnant rats only presented signs of body weight gain and ungroomed fur. The pregnant rats induced by HFHS combined with MR only showed signs of listlessness and inactivity.

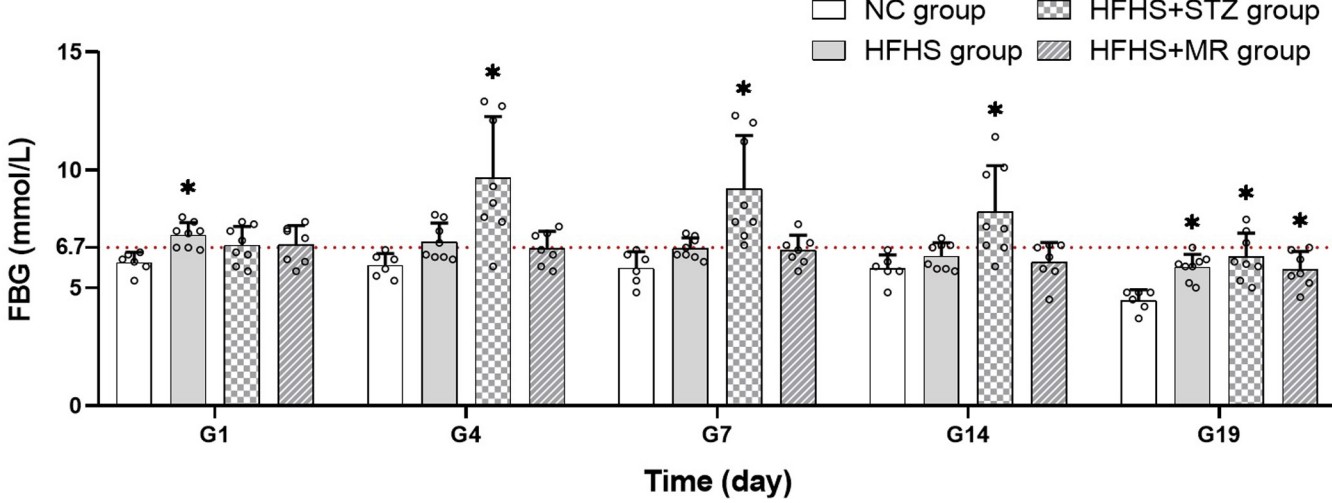

**Fig 1. FBG of the four rat groups.** G: gestational days (day). $^*P < 0.05$ vs. the NC group.

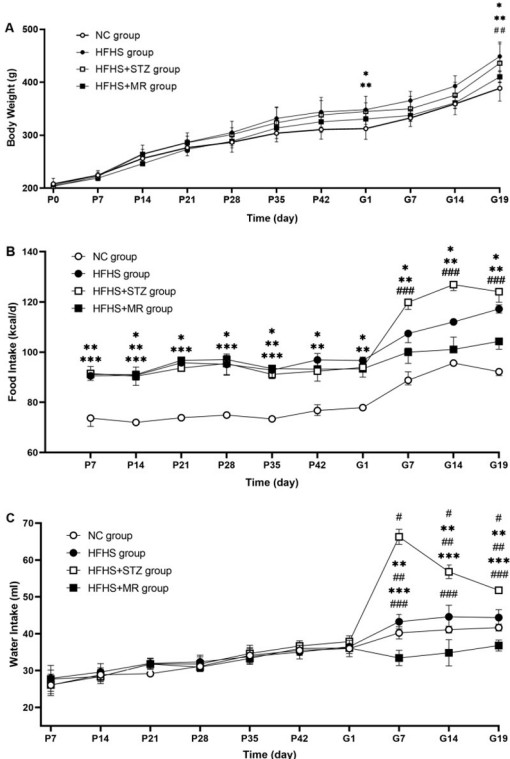

**Fig 2.** (A) Body weight, (B) food intake, and (C) water intake of the four rat groups. P: pre-pregnant days (d); G: gestational days (d). $*P < 0.05$ HFHS group vs. NC group; $**P < 0.05$ HFHS+STZ group vs. NC group; $***P < 0.05$ HFHS+MR group vs. NC group; $\#P < 0.05$ HFHS+STZ group vs. HFHS group; $\#\#P < 0.05$ HFHS+MR group vs. HFHS group; $\#\#\#P < 0.05$ HFHS+STZ group vs. HFHS+MR group.

## Change in FINS, HOMA-IR, HOMA-IS, and HOMA-β

Compared with the rats in the NC group, the FINS and HOMA-IR of the rats in the HFHS, HFHS+STZ, and HFHS+MR groups had significantly increased on the 19th day of pregnancy (Fig 3A and 3B), whereas HOMA-IS had decreased (Fig 3C). These results indicated that the three modeling methods could induce the GDM symptoms of IR and reduce insulin sensitivity. In addition, the HOMA-β of rats in the three modeling groups did not significantly differ from that of the rats in the NC group (Fig 3D).

## Structural and morphological changes in pancreatic and placental tissues

The pancreatic islet cells of the rats in the NC group were neatly aligned and had complete structures and uniform sizes. The means of horizontal diameter and vertical diameter of the islet cells in the NC group were respectively 0.11 mm and 0.10 mm, and the average number of islet cells in per area of pancreatic tissue was 1.67 (Fig 4A). The pancreatic islet cells of the rats in the HFHS group had increased in number and showed irregular arrangements and shapes. The means of horizontal diameter and vertical diameter of the islet cells in the HFHS group were respectively 0.09 mm and 0.13 mm, and the average number of islet cells in per area of pancreatic tissue was 3 (Fig 4B). The pancreatic islet cells in the HFHS+STZ group had decreased in number and had significantly atrophied. The means of horizontal diameter and vertical diameter of the islet cells in the HFHS+STZ group were respectively 0.06 mm and 0.06 mm, and the average number of islet cells in per area of pancreatic tissue was 1 (Fig 4C). The

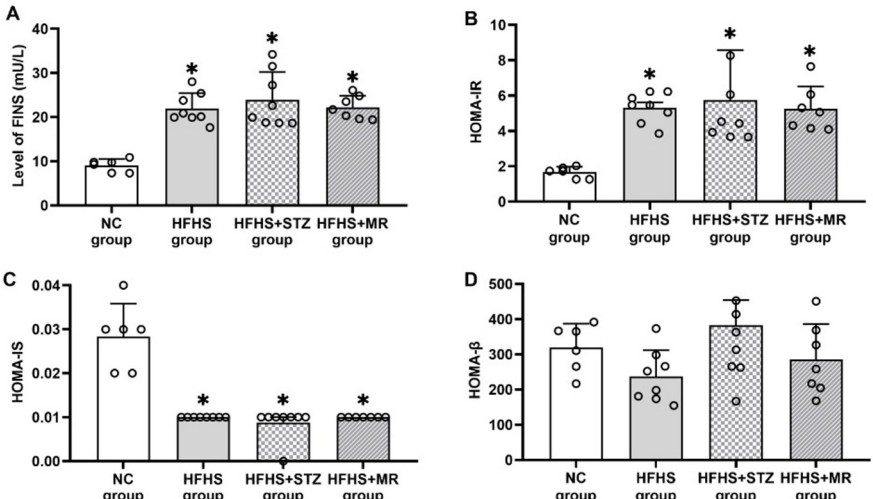

**Fig 3.** (A) FINS, (B) HOMA-IR, (C) HOMA-IS, and (D) HOMA-β of the four rat groups. $^*P < 0.05$ vs. the NC group.

pancreatic islet cells in the HFHS+MR group had decreased in number and had atrophied. The means of horizontal diameter and vertical diameter of the islet cells in the HFHS+STZ group were respectively 0.06 mm and 0.09 mm, and the average number of islet cells in per area of pancreatic tissue was 1 (Fig 4D).

The placental tissue of the rats in the NC group had clear boundaries, neat edges, and uniform cell distribution, gap sizes, and capillary distribution. The averages of the gap area and the vascular distribution area were respectively 0.02 mm$^2$ and 0.05 mm$^2$ (Fig 5A). The placental tissue of the rats in the HFHS group had irregularly stratified boundaries, loose cell distribution, and enlarged gaps. The averages of the gap area and the vascular distribution area were respectively 0.07 mm$^2$ and 0.07 mm$^2$ (Fig 5B). The placental tissue of the rats in the HFHS+STZ group presented disordered stratification. In this group, the trophoblast cells exhibited high vacuolation, and the capillaries had increased excessively and were unevenly distributed. The averages of the gap area and the vascular distribution area were respectively 0.11 mm$^2$ and 0.09 mm$^2$ (Fig 5C). The placental tissue of the rats in the HFHS+MR group had irregularly stratified boundaries, and increased intercellular spaces and capillaries with dispersed distribution. The averages of the gap area and the vascular distribution area were respectively 0.05 mm$^2$ and 0.06 mm$^2$ (Fig 5D).

## Protein expression levels of GLUT1 and GLUT3

The expression levels of GLUT1 and GLUT3 in the placentas of the rats in the HFHS+STZ and HFHS+MR groups were higher than those in the NC and HFHS groups (Fig 6A and 6B).

## Placental proteomic analysis of the HFHD+STZ and NC groups

Proteomics analysis generated 191 288 total spectra corresponding to 28 325 peptides and 4854 proteins, among which 4051 were quantifiable (Fig 7A). The lengths of the identified peptides ranged from 7 aa to 20 aa (Fig 7B). The molecular masses of most of the proteins ranged from 10 kDa to 100 kDa (Fig 7C). Comparing the HFHD+STZ group with the NC group revealed that a total of 23 different proteins with more than 1.3-fold change in expression, including five up-regulated and 18 down-regulated proteins, had been successfully identified

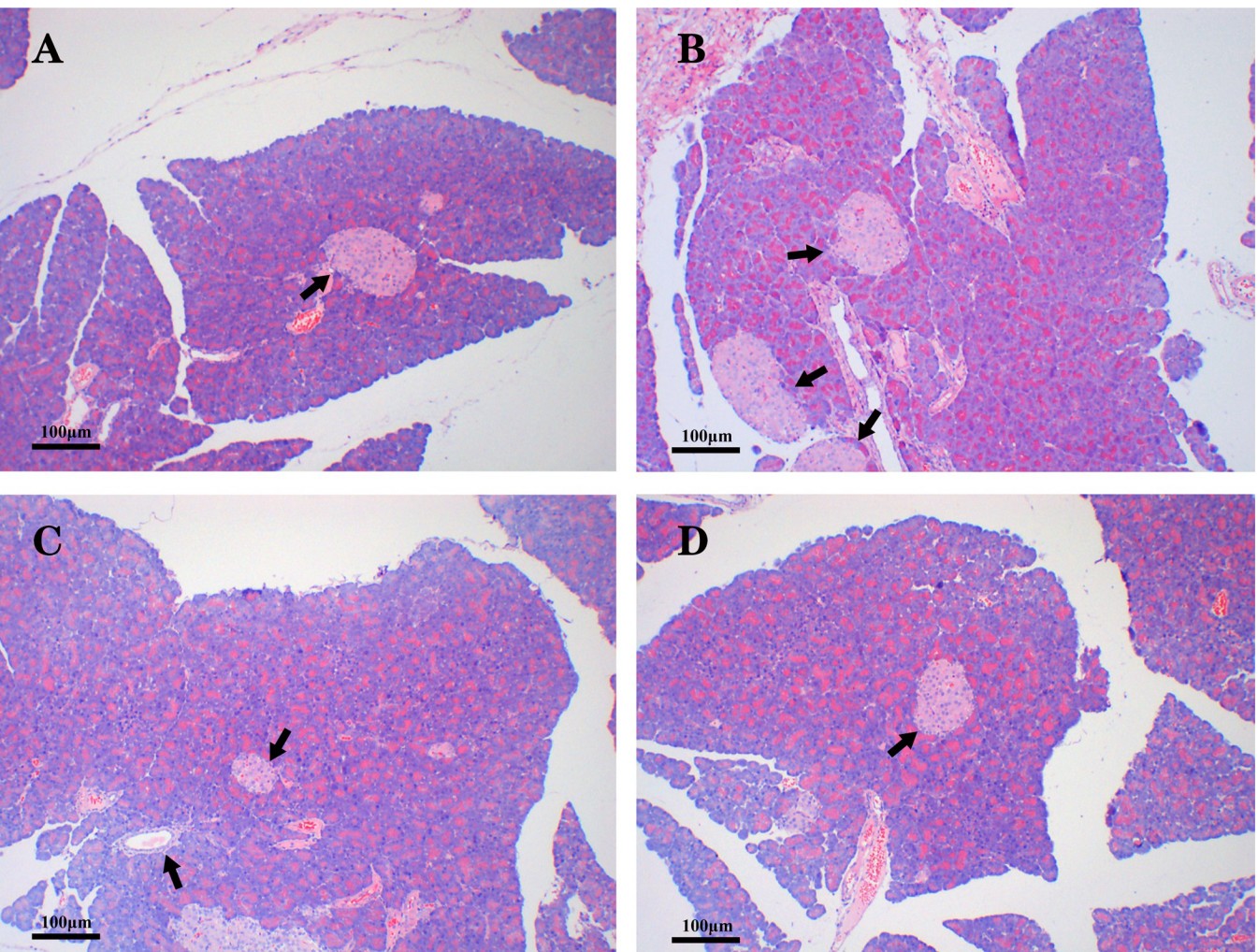

**Fig 4. Structural and morphological changes in islet cells in the four rat groups.** HE-stained pancreatic tissue sections from (A) the NC group, (B) the HFHS group, (C) the HFHS+STZ group, and (D) the HFHS+MR group. Black arrows: islet β cells. Scale bar = 100 μm.

(Table 1). The volcano map shows that the differentially expressed proteins were distributed into two different quadrants (Fig 7D).

## Bioinformatics analysis of differentially expressed proteins in the placenta

GO enrichment divided the differentially expressed proteins into three non-overlapping categories, namely, BP: sensory perception of pain, cellular response to UV and cellular response to corticosteroid stimulus, CC: symbiont-containing vacuole, host cell cytoplasm part, and host cell cytoplasm, and MF: sulfur compound binding, heparin binding, and glycosaminoglycan binding (Fig 8A). KEGG pathway analysis indicated that Keratin 28 and Heat-shock 70 kDa protein 1-like were the critical signaling molecules enriched in the estrogen signaling pathway (Fig 8B).

## Discussion

This work showed that the GDM rat models induced by the HFHS diet combined with 25mg/kg STZ injection had the highest modeling rate and that their FBG during pregnancy was

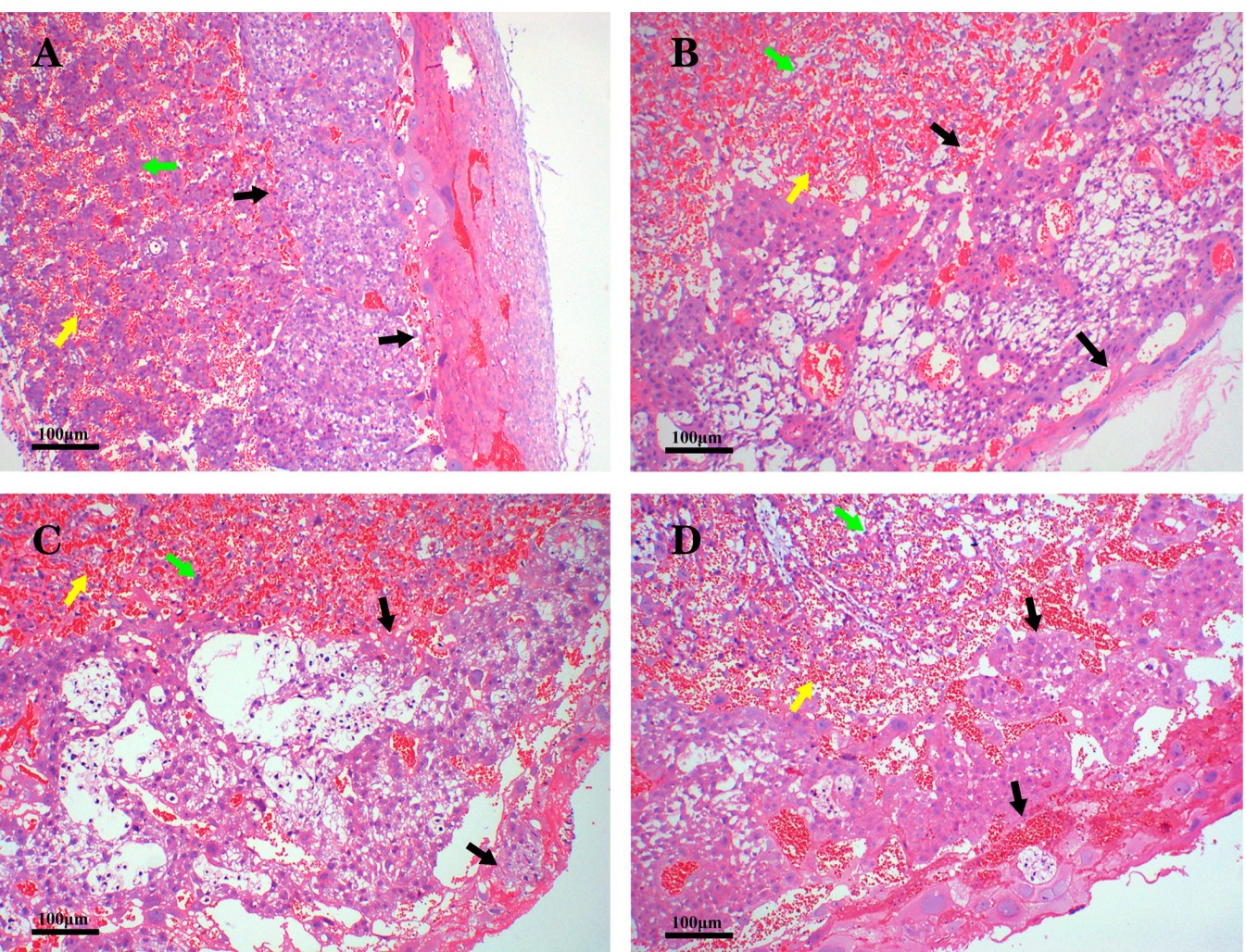

**Fig 5. Structural and morphological changes in the placental tissue in the four rat groups.** HE-stained placental tissue sections from (A) the NC group, (B) the HFHS group, (C) the HFHS+STZ group, and (D) the HFHS+MR group. Black arrows: placental stratification boundary. Yellow arrows: capillaries. Green arrows: trophoblast cells. Scale bar = 100 μm.

maintained in a moderately hyperglycemic status. The rats in the HFHS+STZ group showed the typical GDM characteristics of polydipsia, polyphagia, polyuria, high body weight, ungroomed fur, and inactivity. These features were accompanied by obvious IR and decreased insulin sensitivity and the other pathological manifestations of diabetes, such as severe pancreatic and placental tissue morphology disorders and vacuolization. HFHS induction and HFHS combined with MR induction had a low modeling rate. Although the rat models induced through these methods showed decreased IR and insulin sensitivity during pregnancy and mild to moderate pancreatic and placental pathological changes, their FBG was unstable. No typical clinical signs of GDM were observed. Moreover, Western blot analysis revealed that the expression levels of GLUT1 and GLUT3 increased only in the placentas of the HFHS+STZ and HFHS+MR rat groups. The above results showed that the HFHS+STZ rat group had the best performance in the modeling indicators. Then, proteomics revealed that 23 proteins related to GDM and its complications were differentially expressed in the placentas of the HFHS+STZ rat group. These differentially expressed proteins (DEPs) were enriched in the estrogen signaling pathway.

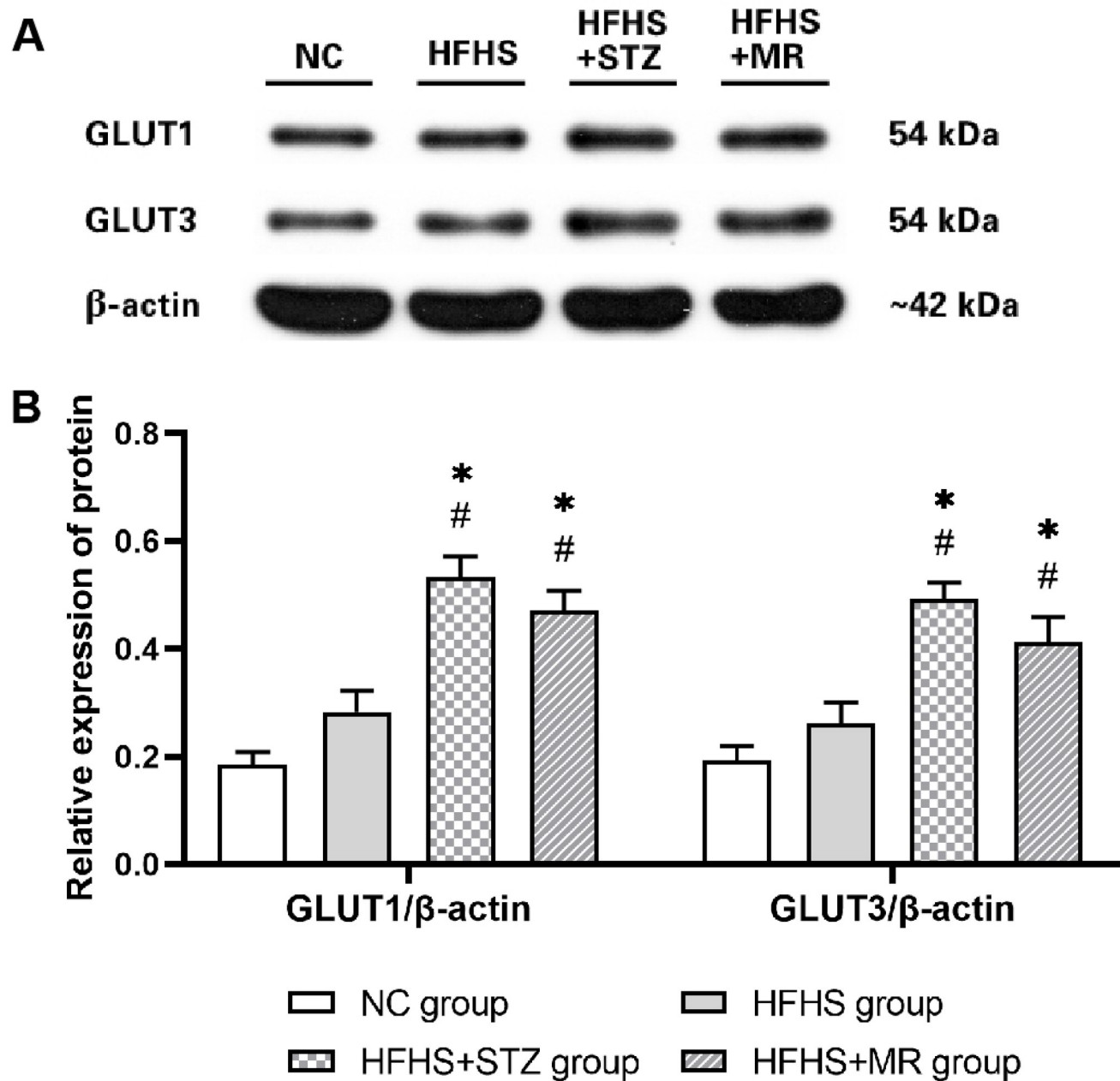

**Fig 6. Protein expression levels of GLUT1 and GLUT3 in the four rat groups.** (A) Effects on the protein expression levels of GLUT1 and GLUT3 and the internal reference β-actin as determined by Western blot analysis, (B) Comparison of the protein expression levels of GLUT1 and GLUT3. * $P < 0.05$ vs. the NC group; # $P < 0.05$ vs. the HFHS group.

GDM is generally believed to be related to the impairment of glucose tolerance caused by pancreatic β-cell dysfunction against the background of chronic IR [20]. Epidemiological studies have identified numerous risk factors for GDM, including dietary changes during pregnancy, body weight gain, reduced activity, and physical and mental stress caused by changes in the social environment [21]; each of these risk factors is directly or indirectly related to impaired pancreatic β-cell function and/or insulin sensitivity. Therefore, an experimental animal model that aims to mimic the pathogenesis and clinical features of human GDM should be constructed on the basis of the pathogenic factors and pathogenesis of GDM.

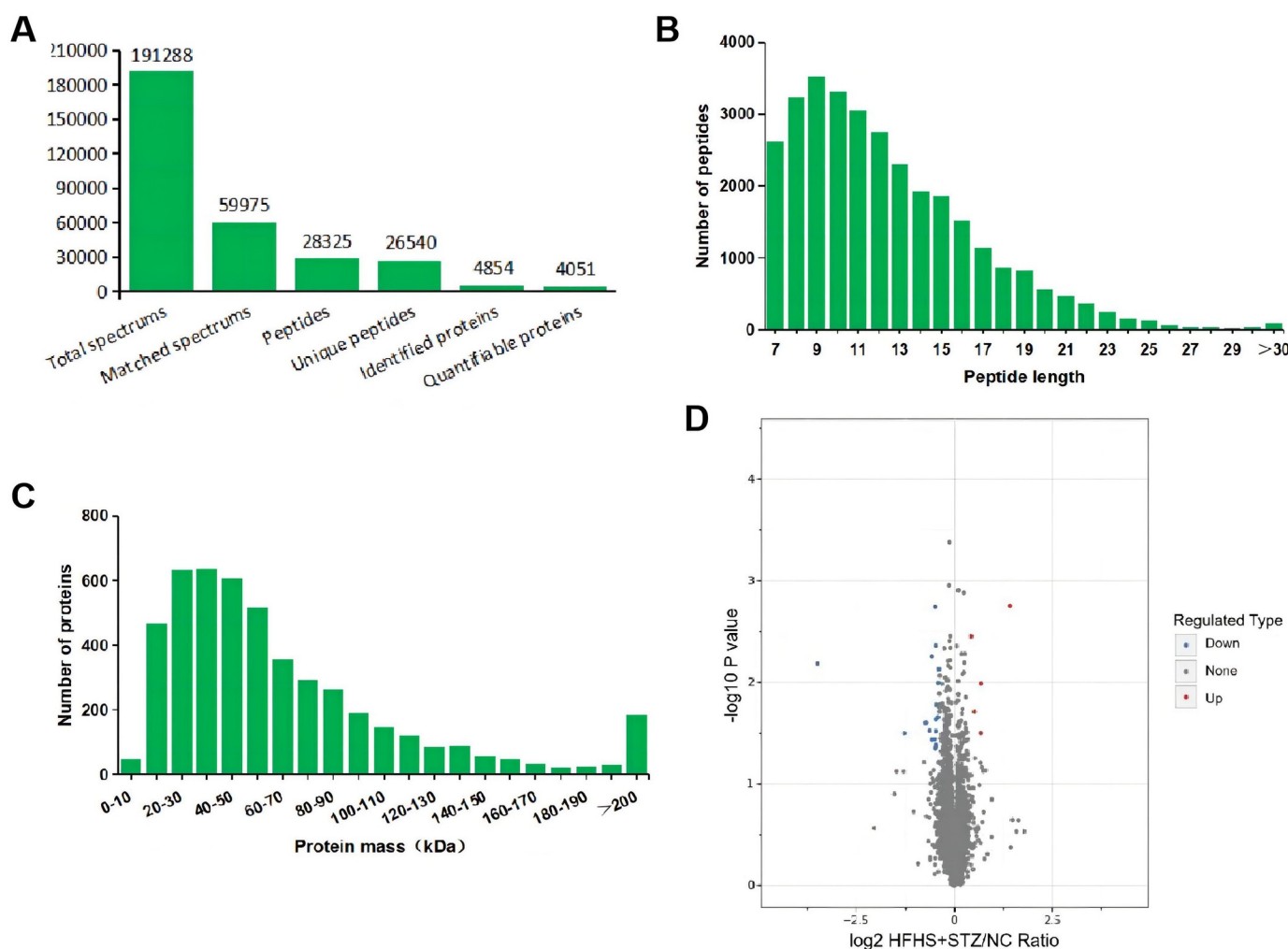

**Fig 7. Overview of differentially expressed proteins and peptides.** (A) Basic statistics of mass spectral data, (B) Peptide length, (C) Protein mass, and (D) Volcano plot of differentially expressed placental proteins between the HFHS+STZ and NC groups.

In recent years, the HFHS diet has been widely used to construct GDM rat models. The modified HFHS diet used in our research was composed of 66.5% basic feed, 20% sucrose, 10% cooked lard, 2.5% cholesterol, and 1% bile salt. Among these components, sucrose and fat are essential factors for inducing hyperglycemia and IR in rats and are added at the proportion of less than 20% to ensure that the food intake and nutritional balance of rats are normal [8]. The presence of 2.5% cholesterol in the diet can not only lead to systemic inflammation and IR [22] but also prevents the death of rats from diarrhea caused by indigestion [8]. In addition, the addition of 1% bile salt can promote cholesterol digestion and increase the risk of GDM without affecting the appetite of rats [8, 23]. Our findings showed that the HFHS diet prepared in accordance with the above formula could induce a GDM rat model with a success rate of 50%. Low-dose STZ is often used to assist in constructing the experimental rat models of GDM [11]. In pregnant rats fed with the HFHS diet, a single low-dose intraperitoneal injection of STZ can mildly impair the insulin secretion function of pancreatic islet β cells and effectively induce hyperglycemia and form pathogenesis similar to GDM [14, 15]. Consistent with other studies, this study demonstrated that the rat HFHS diet combined with a single injection of 25

**Table 1.  Differentially expressed proteins in the placentas of the HFHS+STZ and NC groups.**

| Protein accession | Protein description | Gene name | P value | Regulated |
|---|---|---|---|---|
| A0A0H2UHQ9 | Synaptopodin | Synpo | 0.01032 | Down |
| A0A140TAE2 | RCG31730, isoform CRA_a | LOC500956 | 0.03687 | Down |
| B0BNJ4 | ETHE1, persulfide dioxygenase | Ethe1 | 0.00442 | Down |
| D3ZVB0 | AP-5 complex subunit beta-1 | Ap5b1 | 0.00184 | Down |
| D4AEG5 | RNA helicase | Dhx35 | 0.00668 | Down |
| D3ZPF2 | Malonyl-CoA-acyl carrier protein transacylase | Mcat | 0.00361 | Up |
| D4A7B6 | Transmembrane protein 87B | Tmem87b | 0.01044 | Up |
| F1LVX2 | EH domain-binding protein 1 | Ehbp1 | 0.02359 | Down |
| F1LPV2 | Very low-density lipoprotein receptor | Vldlr | 0.04171 | Down |
| G3V7V5 | Peptidylprolyl isomerase | Fkbp11 | 0.04352 | Down |
| M0RBK3 | Zinc finger protein 36, C3H1 type-like 2-like | LOC100911319 | 0.00755 | Down |
| P07861 | Neprilysin | Mme | 0.03715 | Down |
| P29975 | Aquaporin-1 | Aqp1 | 0.03059 | Down |
| P55063 | Heat shock 70 kDa protein 1-like | Hspa1l | 0.03212 | Down |
| P06765 | Platelet factor 4 | Pf4 | 0.03203 | Up |
| Q587K3 | Potential RabGAP | Tbc1d10a | 0.04506 | Down |
| Q5M7T9 | Threonine synthase-like 2 | Thnsl2 | 0.00564 | Down |
| Q5U202 | Csrp2 protein | Csrp2 | 0.02247 | Down |
| Q5U220 | Transmembrane protein 254 | Tmem254 | 0.01685 | Down |
| Q6IFW7 | Keratin 28 | Krt28 | 0.02551 | Down |
| Q9EQP5 | Prolargin | Prelp | 0.03023 | Down |
| Q9JKB7 | Guanine deaminase | Gda | 0.01990 | Up |
| Q4FZZ3 | Glutathione S-transferase alpha-5 | Gsta5 | 0.00180 | Up |

mg/kg STZ into the intraperitoneal cavity yielded a GDM modeling rate of 87.5% and exhibited a short modeling period and good model stability [24, 25].

In addition, providing HFHS diet rats with a space that can meet only their needs for movement, rest, and normal postural adjustments can induce the stress experienced by patients with GDM that is caused by changes in dietary structure, reduced activity, and anxiety about the prognosis of the disease [17, 26]. Studies have demonstrated that pregnant women given a HFHS diet and are exposed to small spaces for a long time suffer from severe physical and mental stress and present significantly increased blood sugar and IR levels [12]. A study by Reidy and colleagues showed that the development of IR induced by reducing rodent activity through reducing cage size tended to be faster and more pronounced than that of IR induced by a high-fat diet [27]. An animal experiment has shown that reducing the movement space in mice to simulate a lack of physical activity can lead to hyperinsulinemia, muscle IR, dysglycemia and fat gain [28]. This study showed that the HFHS diet combined with MR could be used to induce GDM rat models with the modeling rate of 41.9%.

This study revealed that the FBG levels of pregnant rats in the three modeling groups increased by varying degrees, among which the FBG levels of the pregnant rats in the HFHS +STZ group showed the greatest increment and were maintained in a moderately high blood sugar state during pregnancy. The absence of a significant increase in blood glucose levels in one rat in the HFHS+STZ group might be caused by individual differences in its reaction to STZ [29]. The FBG of the rats in the HFHS group from the fourth day to the 14th day of pregnancy and that of the rats in the HFHS+MR group from the first day to the 14th day of pregnancy did not increase significantly, and that of the three groups gradually decreased during

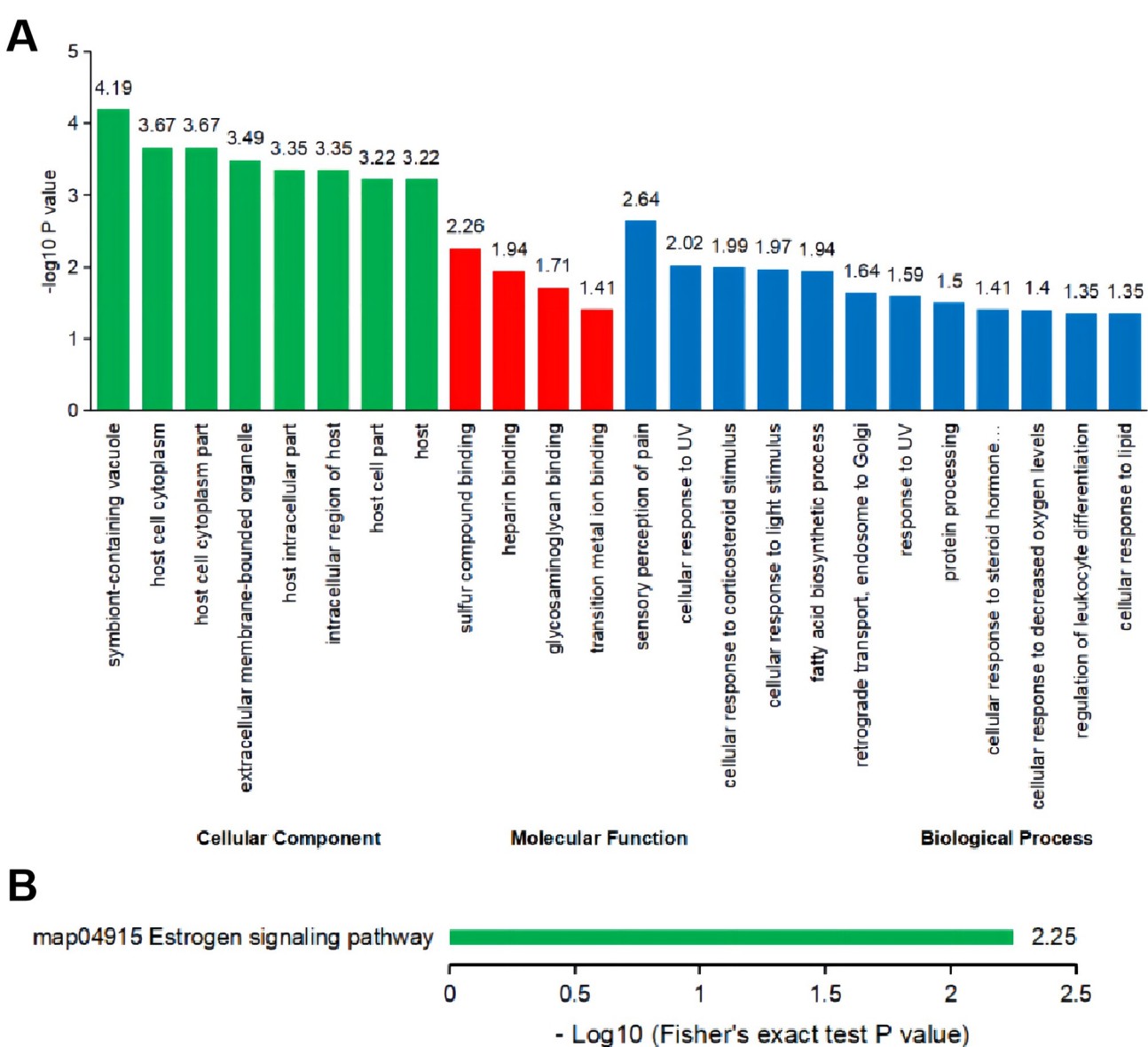

**Fig 8. Bioinformatics analysis of placental proteomics.** (A) GO enrichment of the differentially expressed proteins was obtained by using Fisher's exact test. (B) KEGG pathway enrichment of the differentially expressed proteins.

pregnancy likely because of the strong compensatory capability and high self-healing power of the rats [30].

Moreover, among the three groups, the rats in the HFHS+STZ group presented the most obvious clinical symptoms of GDM with polydipsia, polyphagia, and body weight gain, and high IR level and low insulin sensitivity. Although the rats in the HFHS group had polyphagia, and increased body weight and IR levels, there was no significant change in water intake. The study has shown that reducing water intake deteriorated glycemic regulation, and the high-fat and high-sugar diet in our study only caused mild hyperglycemia in rats on the first and 19th day of gestation, which may be related to the lack of significant changes in water intake during pregnancy [31]. The increased FINS and IR levels, reduced insulin sensitivity, no significant increase in body weight and food intake, and significantly decreased water intake of the rats in

the HFHS+MR group during pregnancy may be attributed to the limited activity and psychological pressure caused by activity restriction [32].

The pancreas is a vital organ that secretes insulin to regulate blood sugar during pregnancy [33]. This study showed that the islet cells proliferated and hypertrophied in the HFHS group, were decreased in number and slightly atrophic in the HFHS+MR group, and were severely atrophic in the HFHS+STZ group. The above can explain why the HFHS group had a high modeling rate but showed no significant increase in FBG; why the HFHS +STZ group had a high modeling rate and its FBG was stably maintained in a moderately hyperglycemic state; and why the HFHS+MR group had a low modeling rate and its FBG did not increase significantly. In addition, the literature has shown that well-controlled glucose levels during pregnancy are usually related to normal placental morphology [34]. The results of this work showed that the placental tissues stratification of the NC rats is clear, the cells and capillaries are evenly distributed, and the gaps are the same size, whereas that of the GDM rats had an obviously disordered hierarchical structure and increased intercellular space and blood vessels at the maternal-fetal interface. The HFHS+STZ group showed more severe placental tissue damages than the other two groups and obvious vacuolization and excessive capillary formation.

The placenta is the only interface between the mother and the fetus and is the organ for exchanging gases and nutrients between the two. Given that it also has an endocrine function and can express almost all cytokines, it has become a key target in studies on the pathogenesis of GDM [35]. Our previous research discovered that several DEPs are related to various biological processes in the placenta of patients and rats with GDM [36, 37], and that abnormal protein expression in the placenta can mediate insulin antagonism, thus causing IR and abnormal glucose metabolism [38]. Therefore, the expression levels of placental proteins can further determine the stability and reliability of GDM rat models. Previous studies have demonstrated that the expression and activity of specific GLUTs in the placentas of patients with GDM are the main regulatory factors in the process of maternal-fetal glucose exchange [39]. GLUT1 and GLUT3 are the primary glucose transporters in placental cells such as syncytiotrophoblast, cytotrophoblast and vascular endothelial cells [40]. Similar to those in previous reports, the expression levels of GLUT1 and GLUT3 in the placentas of the HFHS+STZ and HFHS+MR groups were increased in this study [41]. Studies have shown that hyperglycemia can increase the expression levels of GLUT1 and GLUT3 protein in the placenta [42, 43], which may be likely relevant to the increased FBG, the increased capillary density and trophoblast disorder in the placental tissue of the GDM rat models. By contrast, GLUT protein expression levels in the placentas of the rats in the HFHS group did not change significantly. We speculated that this result might be caused by the effect of the nutritional hormone signals of the placenta on the proliferation and hypertrophy of pancreatic β cells during pregnancy that drives the compensatory adaptation of the pancreatic endocrine system to maintain normal blood sugar levels [44].

The present data showed that the GDM rat experimental model induced through the HFHS diet combined with a single intraperitoneal injection of 25 mg/kg STZ was the most successful and stable. Therefore, we conducted iTRAQ proteomic analysis on the placentas of the rats in the HFHS+STZ and NC groups. Comparison with the NC group revealed five up-regulated and 18 down-regulated proteins in the placentas of the rats in the HFHS+STZ group. Among the identified proteins, five DEPs including very low-density lipoprotein receptor (Vldlr), aquaporin-1 (Aqp1), platelet factor 4 (Pf4), peptidylprolyl isomerase and malonyl-coa-acyl carrier protein transacylase (Mcat), have been confirmed to be related to the change in placental function, placental vascular dysfunction, and placental inflammation in patients with GDM and its complications [45–50]. Decreased levels of Vldlr may

contribute to the development of GDM by inhibiting the ability of the placenta to clear cholesterol [45]. Aqp1 plays a key role in the maintenance of amniotic fluid homeostasis, and its expression level can be decreased with the deepening of insulin resistance during pregnancy, which may explain the frequent abnormal amniotic fluid homeostasis in pregnant women with diabetes [46]. Pf4 may be a new marker for monitoring coagulation function in patients with GDM and may also be a risk factor for GDM during pregnancy [47, 48]. Lanoix et al. [49] found that peptidylprolyl isomerase can be used as a protein reference marker for GDM placental research. Mcat may indirectly trigger GDM by participating in fatty acid biosynthesis during pregnancy [50]. These findings further proved that the HFHS diet combined with a single intraperitoneal injection of 25 mg/kg STZ could induce molecular biological changes that are similar to the changes observed in human patients with GDM [51] and can thus increase the reliability of the GDM rat model. Moreover, we found that the DEPs in the placenta were enriched in the estrogen signaling pathway. Estrogen can regulate the quality and function of pancreatic β cells to adapt to physiological IR during pregnancy by working together with the estrogen receptor and G protein-coupled estrogen receptor and by protecting β cells from apoptosis caused by high glucose stimulation [44, 52]. Pathological IR may be induced if estrogen functions in an inappropriate stage of pregnancy or if its level are not within the physiological range [52].

## Conclusions

Our findings indicated that the modified HFHS diet combined with a single intraperitoneal injection of 25 mg/kg STZ was the optimal method for constructing a rat model of nonspontaneous GDM. Compared with the other two models, the rat model of the HFHS+STZ group had the advantage of better representing the broad phenotype and pathology of human GDM, and the model featured high stability. Compared with the HFHS+STZ modeling method, the pathogeny simulation of the HFHS+MR group were more similar to the etiology of human GDM and also caused obvious pathological changes in rats, but the modeling rate was lower than that of other methods. In follow-up research, we will modify the method of movement restriction and physical activity measure, such as using telemetry devices, in order to construct a new rat model that better match the characteristics of human GDM.

## Supporting information

**S1 File.**
(DOCX)

**S2 File.**
(XLSX)

**S1 Text.**
(TXT)

## Acknowledgments

We extend our gratitude to the animal care staff of the Animal Experiment Center of Fujian University of Traditional Chinese Medicine for their support and assistance in the maintenance and feeding of rats. We also thank all the participants of this study.

## Author Contributions

**Conceptualization:** Li Ge, Xinyong Jiang, Yuting Lai, Pingping Huang, Xiumin Jiang.

**Data curation:** Fan Chen, Xinyong Jiang, Yuting Lai, Yuzheng Lin.

**Formal analysis:** Fan Chen, Yuting Lai.

**Funding acquisition:** Li Ge.

**Investigation:** Fan Chen, Xinyong Jiang, Yuting Lai.

**Methodology:** Fan Chen, Li Ge, Xinyong Jiang, Yuting Lai, Jinghe Hua.

**Project administration:** Li Ge, Pingping Huang, Xiumin Jiang.

**Resources:** Fan Chen, Li Ge.

**Software:** Fan Chen, Xinyong Jiang.

**Visualization:** Fan Chen, Xinyong Jiang, Yan Lin.

**Writing – original draft:** Fan Chen, Li Ge, Xinyong Jiang.

**Writing – review & editing:** Fan Chen, Li Ge, Jinghe Hua.

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
