## [Decision Letter · Decision Letter 0]

21 Mar 2022

PONE-D-21-38566Construction of the experimental rat model of gestational diabetesPLOS ONE

Dear Dr. Ge,

Thank you for submitting your manuscript to PLOS ONE. After careful consideration, we feel that it has merit but does not fully meet PLOS ONE’s publication criteria as it currently stands. Therefore, we invite you to submit a revised version of the manuscript that addresses the points raised during the review process.

We look forward to receiving your revised manuscript.

Kind regards,

Michael Bader

Academic Editor

PLOS ONE

Journal Requirements:

2. To comply with PLOS ONE submissions requirements, in your Methods section, please provide additional information regarding the experiments involving animals and ensure you have included details on (1) methods of anesthesia and/or analgesia, and (2) efforts to alleviate suffering.

In your cover letter, please note whether your blot/gel image data are in Supporting Information or posted at a public data repository, provide the repository URL if relevant, and provide specific details as to which raw blot/gel images, if any, are not available. Email us at plosone@plos.org if you have any questions

Reviewers' comments:

Reviewer's Responses to Questions

**Comments to the Author**

1. Is the manuscript technically sound, and do the data support the conclusions?

Reviewer #1: Yes

Reviewer #2: Yes

2. Has the statistical analysis been performed appropriately and rigorously? 

Reviewer #1: Yes

Reviewer #2: No

3. Have the authors made all data underlying the findings in their manuscript fully available?

Reviewer #1: No

Reviewer #2: Yes

4. Is the manuscript presented in an intelligible fashion and written in standard English?

Reviewer #1: Yes

Reviewer #2: Yes

5. Review Comments to the Author

Reviewer #1: Present work deeply explores metabolic, biochemical and histopathologic changes produced by three different kinds of situation added to hypercaloric diet administration for 6 weeks previous to pregnancy and during gestation in rats. The hypothesis and methodology is clearly explained, and the variables compared are multiple.

Results point out towards achieving a slight better experimental Gestational Diabetes Mellitus model when animals were treated with hypercaloric diet plus a single low dose of streptozotocin compared with diet alone and diet with movement restriction. Authors conclude the latter combination is the best model of gestational mellitus.

I think this paper makes a significant contribution to the elucidation of gestational diabetes mellitus pathophysiology and I will recommend only to discuss more deeply the similarity to human conditions that trigger GDM which not always include a severe condition that can be represented represented by the other experimental models. Also, the plausible role of the findings on the placental GLUT transporters can also be discussed.

Reviewer #2: In this article, the authors compare commonly used rat models of gestational diabetes (HFHS and HFHS+STZ) with a model that adds movement restriction to the HFHS model. The authors include various physiological and morphological characterizations of these three models, compared to females maintained on standard chow, and conclude that the HFHS+STZ model is the best suited to model GDM. The study concludes with a proteomic analysis of placenta from chow-fed and HFHS+STZ rat dams.

I had several minor and major concerns that I've itemized below.

Major Concerns:

1. Analysis of the pancreas should be more systematic. The number of islets per area of tissue should be calculated, and perform measures of islet diameter and sphericity in a blinded manner. At present, the mere descriptions of the images without quantification are difficult to interpret.

2. Similar for the placenta analysis. I think a quantification or at least well-described semi-quantification of the various factors described (cell distribution, gap size, trophoblast count, vacuolation area, capillary number and distribution) would make these data more meaningful and reproduceable.

3. Please provide the caloric density of the chow and HFHS (as kcal/g or KJ/g) in the methods, and then present the food intake data in these units, rather than in g. Assuming that the diets are not equicaloric, the caloric intake is more relevant that the number of grams consumed. This conversion will likely change the results and interpretation of the data shown in Figure 3. For example, assuming that that the HFHS diet is more calorically dense, then the HFHS+STZ group is likely consuming more calories during pregnancy than the NC group, the NC and HFHS groups are consuming similar calories, and the HFHS+MR are consuming even fewer calories than represented in grams. Of course the amount of calories consumed by these groups would have consequences for the growing fetuses and the maternal system.

4. There is no discussion about the female rats that failed to show GDM under the various models of induction. I actually think these sub-populations, those that seem resilient in the face of the diet, chemical, and mobility challenges, would be interesting to discuss and study further.

Minor Concerns:

1. In line 71, you mentioned gene-induced spontaneous models of GDM. It would be interesting to tell the reader the mutation of which genes lead to these models.

2. Please provide a citation for the use of 6.7 mmol/L on the fourth day of pregnancy for the criteria for GDM.

3. How activity was reduced in the MR group is not fully clear. What exactly is a baffle? Can you better define the percent reduction in cage floor area in the MR group? Did the barrier actually prevent the females from accessing food and water, or just make it more difficult to access? In future studies, measurement of physical activity with telemetry devices would be useful for thoroughly quantifying the reduction in movement in this model.

4. Please provide references for the HOMA calculations.

5. Considering using “unkept or ungroomed fur” in the place of “messy hair.”

6. Please include individual data points in Figure 3 graphs.

7. Please confirm, in Figure 6B, GLUT1 and GLUT3 expression are presented relative to B-actin expression. Define this more clearly.

8. Upon adjusting the food intake to caloric density, be sure to modify the discussion of these results (Lines 488-489).

9. A study by Boileau and colleagues (PMID: 7615800) showed that hyperglycemia can stimulate GLUT3 protein levels in the placenta. This finding is likely relevant given the increase FBG in the HFHS+STZ group.

10. I would appreciate a more detailed discission of the change in placenta protein expression profile. Which five DEPs are related to placental function? How might this information be useful for understanding GDM etiology or for the development of prevention or treatment strategies?

11. In the conclusion, it is stated that “The characteristics induced by the HFHS+MR modeling method were more in line with the pathological characteristics of GDM than those induced by the other modeling methods,” but in Lines 483-484, the authors state that, “only the rats in the HFHS+STZ group presented the obvious clinical manifestations of GDM with polydipsia, polyphagia, polyuria, and body weight gain and the typical pathogenesis of GDM with high IR levels and decreased insulin sensitivity.” For me these two statements do not align with each other. Perhaps the authors are making the point that the cause of the pathology in the HFHS+MR group is more similar to the etiology of human GDM, compared to the STZ-induced model? Please make this point clearer.

6. PLOS authors have the option to publish the peer review history of their article (what does this mean?). If published, this will include your full peer review and any attached files.

Reviewer #1: No

Reviewer #2: No

---

## [Author Response · Author response to Decision Letter 0]

23 Jul 2022

Dear Editor and Reviewers:

Thank you so much for your comments concerning our manuscript entitled ‘Construction of the experimental rat model of gestational diabetes’(Manuscript Number：PONE-D-21-38566). These comments are all valuable and very helpful to the revision and improvement of our manuscript. We have read the comments carefully and have made corrections according to these comments. Based on the instructions provided in your letter, we have uploaded the file of the revised manuscript. Revisions in the text are highlighted in yellow. The responses to the reviewers' comments are presented follows:

Part A (Editor): 

1. The editor’s comment: Please ensure that your manuscript meets PLOS ONE's style requirements, including those for file naming. 

Our answer: We have chected all the files and revised the errors.

2.The editor’s comment: To comply with PLOS ONE submissions requirements, in your Methods section, please provide additional information regarding the experiments involving animals and ensure you have included details on (1) methods of anesthesia and/or analgesia, and (2) efforts to alleviate suffering.

Our thoughts and corrections: We added additional methods for euthanizing rats. The details are as follows (see the lines 129-132 in the Materials and Methods section):

In order to alleviate rats’ suffering as far as possible, rats were sacrificed under deep anesthesia by intraperitoneal injection with 20% urethane (1 g/kg body weight) by rapid exsanguination through cutting off the abdominal aorta after collecting sufficient abdominal aortic blood.

2. The editor’s comment: PLOS ONE now requires that authors provide the original uncropped and unadjusted images underlying all blot or gel results reported in a submission’s figures or Supporting Information files. In your cover letter, please note whether your blot/gel image data are in Supporting Information or posted at a public data repository, provide the repository URL if relevant, and provide specific details as to which raw blot/gel images. 

Our thoughts and corrections: The blot/gel image data in the study have been in Supporting Information and are posted at a public data repository which is named “figshare”. The repository URL is https://doi.org/10.6084/m9.figshare.19802617.v1 .We provided blot raw images. Please see "S1_File. Original images for blots and gels" for details.

Part B (Reviewer 1):

1. The reviewer’s comment: I think this paper makes a significant contribution to the elucidation of gestational diabetes mellitus pathophysiology and I will recommend only to discuss more deeply the similarity to human conditions that trigger GDM which not always include a severe condition that can be represented represented by the other experimental models. Also, the plausible role of the findings on the placental GLUT transporters can also be discussed.

Our thoughts and corrections: Thank you for your suggestion. We discussed deeply the conditions that trigger GDM in human and the plausible role of the placental GLUT transporter in the Discussion section. The details are as follows (see the lines 472-483 and the lines 536-543 in the Discussion section ):

In addition, providing HFHS diet rats with a space that can meet only their needs for movement, rest, and normal postural adjustments can induce the stress experienced by patients with GDM that is caused by changes in dietary structure, reduced activity, and anxiety about the prognosis of the disease [17,26]. Studies have demonstrated that pregnant women given a HFHS diet and are exposed to small spaces for a long time suffer from severe physical and mental stress and present significantly increased blood sugar and IR levels [12]. A study by Reidy and colleagues showed that the development of IR induced by reducing rodent activity through reducing cage size tended to be faster and more pronounced than that of IR induced by a high-fat diet [27]. An animal experiment has shown that reducing the movement space in mice to simulate a lack of physical activity can lead to hyperinsulinemia, muscle IR, dysglycemia and fat gain [28].

[27]Reidy PT, Monnig JM, Pickering CE, Funai K, Drummond MJ. Preclinical rodent models of physical inactivity-induced muscle insulin resistance: challenges and solutions. J Appl Physiol (1985). 2021 Mar 1;130(3):537-544. doi: 10.1152/japplphysiol.00954.2020. Epub 2020 Dec 24. PMID: 33356986; PMCID: PMC7988796.

[28]Mahmassani ZS, Reidy PT, McKenzie AI, Petrocelli JJ, Matthews O, de Hart NM, Ferrara PJ, O'Connell RM, Funai K, Drummond MJ. Absence of MyD88 from Skeletal Muscle Protects Female Mice from Inactivity-Induced Adiposity and Insulin Resistance. Obesity (Silver Spring). 2020 Apr;28(4):772-782. doi: 10.1002/oby.22759. Epub 2020 Feb 28. PMID: 32108446; PMCID: PMC7093260.

GLUT1 and GLUT3 are the primary glucose transporters in placental cells such as syncytiotrophoblast, cytotrophoblast and vascular endothelial cells [40]. Similar to those in previous reports, the expression levels of GLUT1 and GLUT3 in the placentas of the HFHS+STZ and HFHS+MR groups were increased in this study [41]. Studies have shown that hyperglycemia can increase the expression levels of GLUT1 and GLUT3 protein in the placenta [42,43], which may be likely relevant to the increased FBG, the increased capillary density and trophoblast disorder in the placental tissue of the GDM rat models.

[40]Illsley NP, Baumann MU. Human placental glucose transport in fetoplacental growth and metabolism. Biochim Biophys Acta Mol Basis Dis. 2020 Feb 1;1866(2):165359. doi: 10.1016/j.bbadis.2018.12.010. Epub 2018 Dec 26. PMID: 30593896; PMCID: PMC6594918.

[41]Ma XP, Yang HX. Expression of placental glucose transporters in pregnant women with abnormal glycometabolism.Chinese Journal of Perinatal Medicine. 2008;(06):373-376.

[42]Boileau P, Mrejen C, Girard J, Hauguel-de Mouzon S. Overexpression of GLUT3 placental glucose transporter in diabetic rats. J Clin Invest. 1995 Jul;96(1):309-17. doi: 10.1172/JCI118036. PMID: 7615800; PMCID: PMC185202. 

[43]Stanirowski PJ, Szukiewicz D, Majewska A, Wątroba M, Pyzlak M, Bomba-Opoń D, Wielgoś M. Placental expression of glucose transporters GLUT-1, GLUT-3, GLUT-8 and GLUT-12 in pregnancies complicated by gestational and type 1 diabetes mellitus[J]. J Diabetes Investig, 2022,13(3):560-570.

Part C (Reviewer 2):

1. The reviewer’s comment: Analysis of the pancreas should be more systematic. The number of islets per area of tissue should be calculated, and perform measures of islet diameter and sphericity in a blinded manner. At present, the mere descriptions of the images without quantification are difficult to interpret.

Our thoughts and corrections: Thank you for your comments. We supplemented quantitative data for the images of the pancreas. The details are as follows (see the Results section on the lines 343-358):

The pancreatic islet cells of the rats in the NC group were neatly aligned and had complete structures and uniform sizes. The means of horizontal diameter and vertical diameter of the islet cells in the NC group were respectively 0.11 mm and 0.10 mm, and the average number of islet cells in per area of pancreatic tissue was 1.67 (Fig 4A). The pancreatic islet cells of the rats in the HFHS group had increased in number and showed irregular arrangements and shapes. The means of horizontal diameter and vertical diameter of the islet cells in the HFHS group were respectively 0.09 mm and 0.13 mm, and the average number of islet cells in per area of pancreatic tissue was 3 (Fig 4B). The pancreatic islet cells in the HFHS+STZ group had decreased in number and had significantly atrophied . The means of horizontal diameter and vertical diameter of the islet cells in the HFHS+STZ group were respectively 0.06 mm and 0.06 mm, and the average number of islet cells in per area of pancreatic tissue was 1 (Fig 4C). The pancreatic islet cells in the HFHS+MR group had decreased in number and had atrophied. The means of horizontal diameter and vertical diameter of the islet cells in the HFHS+STZ group were respectively 0.06 mm and 0.09 mm, and the average number of islet cells in per area of pancreatic tissue was 1 (Fig 4D). 

2. The reviewer’s comment: Similar for the placenta analysis. I think a quantification or at least well-described semi-quantification of the various factors described (cell distribution, gap size, trophoblast count, vacuolation area, capillary number and distribution) would make these data more meaningful and reproduceable.

Our thoughts and corrections: Thank you for your comments. We supplemented certain quantitative data for the images of the placenta. The details are as follows (see the Results section on the lines 359-372):

The placental tissue of the rats in the NC group had clear boundaries, neat edges, and uniform cell distribution, gap sizes, and capillary distribution. The average gap area and the average vascular distribution area were respectively 0.02 mm2 and 0.05 mm2 (Fig 5A). The placental tissue of the rats in the HFHS group had irregularly stratified boundaries, loose cell distribution, and enlarged gaps. The average gap area and the average vascular distribution area were respectively 0.07 mm2 and 0.07 mm2 (Fig 5B). The placental tissue of the rats in the HFHS+STZ group presented disordered stratification. In this group, the trophoblast cells exhibited high vacuolation, and the capillaries had increased excessively and were unevenly distributed. The average gap area and the average vascular distribution area were respectively 0.11 mm2 and 0.09 mm2 (Fig 5C). The placental tissue of the rats in the HFHS+MR group had irregularly stratified boundaries, and increased intercellular spaces and capillaries with dispersed distribution. The average gap area and the average vascular distribution area were respectively 0.05 mm2 and 0.06 mm2 (Fig 5D). 

3. The reviewer’s comment: Please provide the caloric density of the chow and HFHS (as kcal/g or KJ/g) in the methods, and then present the food intake data in these units, rather than in g. Assuming that the diets are not equicaloric, the caloric intake is more relevant that the number of grams consumed. This conversion will likely change the results and interpretation of the data shown in Figure 3. For example, assuming that that the HFHS diet is more calorically dense, then the HFHS+STZ group is likely consuming more calories during pregnancy than the NC group, the NC and HFHS groups are consuming similar calories, and the HFHS+MR are consuming even fewer calories than represented in grams. Of course the amount of calories consumed by these groups would have consequences for the growing fetuses and the maternal system.

Our thoughts and corrections: Thank you for your comments. We changed the food intake unit to kcal/g, supplemented the calorie content of food in the Methods section, and changed the results and interpretation of the data shown in Figure 3 in the Results section. The details are as follows (see the Method section on the lines 140-145 and Result section on the lines 301-318):

The HFHS diet was composed of 66.5% basic feed, 20.0% sucrose, 10.0% cooked lard, 2.5% cholesterol, and 1.0% cholat, and the total energy obtained from the HFHSD was 4.43 kcal/g in which 34.42%, 12.65%, and 52.93% were derived from fat, protein, and carbohydrates, respectively [8]. The total energy of the standard diet was 3.42 kcal/g in which 12.11%, 22.47%, and 65.42% were derived from fat, protein, and carbohydrates, respectively (GB 14924 1022).

[8]Luo JB, Li JH, Wang HJ, Zhou XY, Zeng T, Zhou J, et al. Metabolic Disease Animal Modelsinduced by High-fat Diets. Laboratory Animal and Comparative Medicine. 2021 Feb;41(01):70-78. Chinese.

Before pregnancy, there was no significant difference in body weight and water intake among the four groups; compared with the rats in the NC group, the food intake of other three groups was significantly increased, but there was no significant difference in the three groups. Compared with the rats in the NC group, the rats in the HFHS group showed increased body weight on the first and 19th days of pregnancy, increased food intake during pregnancy, and no significant changes in water intake. The rats in the HFHS+STZ group displayed ungroomed fur, polyuria, inactivity, increased body weight, food intake and water intake during pregnancy. The rats in the HFHS+MR group exhibited ungroomed fur, listlessness, and inactivity during pregnancy, and decreased water intake on the seventh, 14th, and 19th days of pregnancy, and no significant changes in body weight and food intake. Compared the three modeling groups, the body weight of the rats in the HFHS group was higher than that of the rats in the HFHS+MR group on the 19th day of pregnancy (Fig 2A). On the seventh, 14th, and 19th day of pregnancy, the rats in the HFHS+STZ group had higher food intake than those in the HFHS+MR group (Fig 2B), and had higher water intake than those in the HFHS and HFHS+MR groups; at the same point in time, the rats in the HFHS group had higher water intake than those in the HFHS+MR group (Fig 2C). 

4. The reviewer’s comment: There is no discussion about the female rats that failed to show GDM under the various models of induction. I actually think these sub-populations, those that seem resilient in the face of the diet, chemical, and mobility challenges, would be interesting to discuss and study further.

Our thoughts and corrections: We greatly appreciate your constructive suggestion. Your comments are very great, and we also recognize that it will be very interesting to continue to do a subgroup of female rats that fail to show GDM. We will further explore this subgroup situation in the follow-up study.

5. The reviewer’s comment: In line 71, you mentioned gene-induced spontaneous models of GDM. It would be interesting to tell the reader the mutation of which genes lead to these models.

Our thoughts and corrections: Thank you for your suggestion. We have supplemented the genetically or spontaneous derived GDM models in the background section of this paper. The details are as follows (see the lines 76-78 in the Introduction section):

Current GDM rat models include spontaneous or genetically derived models (BBDP/BBDR rats, Zucker fatty rats, LepRdb rats and Goto-Kakizaki rats) and experiment-induced nonspontaneous models [3]. 

[3]Carrasco-Wong I, Moller A, Giachini FR, Lima VV, Toledo F, Stojanova J, et al. Placental structure in gestational diabetes mellitus. Biochim Biophys Acta Mol Basis Dis. 2020 Feb 1;1866(2):165535. doi: 10.1016/j.bbadis.2019.165535. Epub 2019 Aug 20. PMID: 31442531.

6. The reviewer’s comment: Please provide a citation for the use of 6.7 mmol/L on the fourth day of pregnancy for the criteria for GDM.

Our thoughts and corrections: We applied ref [18] Daikwo M A, Agbana B E, Egene L, James G. The Effect of Aqueous Extract of Gymnema Sylvestre on Blood Sugar Leyel of Alloxan-Induced Diabetic Albino Rats (Rattus Norvergicus). GPH (IJAS)-International Journal of Applied Science. 2020, 3(11): 01-08.

7. The reviewer’s comment: How activity was reduced in the MR group is not fully clear. What exactly is a baffle? Can you better define the percent reduction in cage floor area in the MR group? Did the barrier actually prevent the females from accessing food and water, or just make it more difficult to access? In future studies, measurement of physical activity with telemetry devices would be useful for thoroughly quantifying the reduction in movement in this model.

Our thoughts and corrections: Thanks to your comments. We added the descriptions about the baffles and confinement spaces used by the MR group. Measurement of physical activity with telemetry devices is a great idea which we will used in future studies. The details are as follows (see the lines 161-164 in the Method section):

The activity space of the rats in the HFHS+MR group was reduced 2/3 by placing a baffle made of acrylic plastic in the site of 1/3 cage, where was L: 395 cm × W: 200 cm × H: 200 cm and did not affect the rats normally taking food and water. 

8. The reviewer’s comment: Please provide references for the HOMA calculations.

Our thoughts and corrections: Thanks to your comment, we supplemented the references: [19]Matthews DR, Hosker JP, Rudenski AS, et al.Homeostasis model assessment: insulin resistance and beta-cell function from fasting plasma glucose and insulin concentrations in man. Diabetologia. 1985 Jul;28(7):412-9. doi: 10.1007/BF00280883. PMID: 3899825.

9. The reviewer’s comment: Considering using “unkept or ungroomed fur” in the place of “messy hair.”

Our thoughts and corrections: Thanks to your comment, we replaced all messy hair in this article with ungroomed fur.

10. The reviewer’s comment: Please include individual data points in Figure 3 graphs.

Our thoughts and corrections: Thanks to your comment. We added individual data points in Figure 3 graphs.

11. The reviewer’s comment: Please confirm, in Figure 6B, GLUT1 and GLUT3 expression are presented relative to B-actin expression. Define this more clearly.

Our thoughts and corrections: Thanks to your comment, we have modified the abscissa of Figure 6B in this paper, see Figure 6B for details.

12. The reviewer’s comment: Upon adjusting the food intake to caloric density, be sure to modify the discussion of these results (Lines 488-489).

Our thoughts and corrections: Thanks to your comment, we have revised the Discussion section of the food intake results. The details are as follows (see lines 497-508 in the Discussion section):

Moreover, among the three groups, the rats in the HFHS+STZ group presented the most obvious clinical symptoms of GDM with polydipsia, polyphagia, and body weight gain, and high IR level and low insulin sensitivity. Although the rats in the HFHS group had polyphagia, and increased body weight and IR levels, there was no significant change in water intake. The study has shown that reducing water intake deteriorated glycemic regulation, and the high-fat and high-sugar diet in our study only caused mild hyperglycemia in rats on the first and 19th day of gestation, which may be related to the lack of significant changes in water intake during pregnancy [31]. The increased FINS and IR levels, reduced insulin sensitivity, no significant increase in body weight and food intake, and significantly decreased water intake of the rats in the HFHS+MR group during pregnancy may be attributed to the limited activity and psychological pressure caused by activity restriction [32]. 

[31]Johnson EC, Bardis CN, Jansen LT, Adams JD, Kirkland TW, Kavouras SA. Reduced water intake deteriorates glucose regulation in patients with type 2 diabetes. Nutr Res. 2017 Jul;43:25-32. doi: 10.1016/j.nutres.2017.05.004. Epub 2017 May 17. PMID: 28739050.

[32]Moreno-Fernández S, Garcés-Rimón M, Vera G, Astier J, Landrier JF, Miguel M. High Fat/High Glucose Diet Induces Metabolic Syndrome in an Experimental Rat Model. Nutrients. 2018 Oct 14;10(10):1502. doi: 10.3390/nu10101502. PMID: 30322196; PMCID: PMC6213024.

13. The reviewer’s comment: A study by Boileau and colleagues (PMID: 7615800) showed that hyperglycemia can stimulate GLUT3 protein levels in the placenta. This finding is likely relevant given the increase FBG in the HFHS+STZ group.

Our thoughts and corrections: Thanks to your comment, we had a more in-depth discussion of GLUT transporters. The details are as follows (see the Discussion section on lines 536-543):

GLUT1 and GLUT3 are the primary glucose transporters in placental cells such as syncytiotrophoblast, cytotrophoblast and vascular endothelial cells [40]. Similar to those in previous reports, the expression levels of GLUT1 and GLUT3 in the placentas of the HFHS+STZ and HFHS+MR groups were increased in this study [41]. Studies have shown that hyperglycemia can increase the expression levels of GLUT1 and GLUT3 protein in the placenta [42,43], which may be likely relevant to the increased FBG, the increased capillary density and trophoblast disorder in the placental tissue of the GDM rat models. By contrast, GLUT protein expression levels in the placentas of the rats in the HFHS group did not change significantly. We speculated that this result might be caused by the effect of the nutritional hormone signals of the placenta on the proliferation and hypertrophy of pancreatic β cells during pregnancy that drives the compensatory adaptation of the pancreatic endocrine system to maintain normal blood sugar levels [44].

[40]Illsley NP, Baumann MU. Human placental glucose transport in fetoplacental growth and metabolism. Biochim Biophys Acta Mol Basis Dis. 2020 Feb 1;1866(2):165359. doi: 10.1016/j.bbadis.2018.12.010. Epub 2018 Dec 26. PMID: 30593896; PMCID: PMC6594918.

[41]Ma XP, Yang HX. Expression of placental glucose transporters in pregnant women with abnormal glycometabolism.Chinese Journal of Perinatal Medicine. 2008;(06):373-376.

[42]Boileau P, Mrejen C, Girard J, Hauguel-de Mouzon S. Overexpression of GLUT3 placental glucose transporter in diabetic rats. J Clin Invest. 1995 Jul;96(1):309-17. doi: 10.1172/JCI118036. PMID: 7615800; PMCID: PMC185202. 

[43]Stanirowski PJ, Szukiewicz D, Majewska A, Wątroba M, Pyzlak M, Bomba-Opoń D, Wielgoś M. Placental expression of glucose transporters GLUT-1, GLUT-3, GLUT-8 and GLUT-12 in pregnancies complicated by gestational and type 1 diabetes mellitus[J]. J Diabetes Investig, 2022,13(3):560-570.

13. The reviewer’s comment: I would appreciate a more detailed discission of the change in placenta protein expression profile. Which five DEPs are related to placental function? How might this information be useful for understanding GDM etiology or for the development of prevention or treatment strategies?

Our thoughts and corrections: Thank you for your comment. We additionally discuss the relationship between the five placental differentially expressed proteins that have been identified. The details are as follows (see the lines 554-569 in the Discussion section):

Among the identified proteins, five DEPs including very low-density lipoprotein receptor (Vldlr), aquaporin-1 (Aqp1), platelet factor 4 (Pf4), peptidylprolyl isomerase and malonyl-coa-acyl carrier protein transacylase (Mcat), have been confirmed to be related to the change in placental function, placental vascular dysfunction, and placental inflammation in patients with GDM and its complications [45-50]. Decreased levels of Vldlr may contribute to the development of GDM by inhibiting the ability of the placenta to clear cholesterol [45]. Aqp1 plays a key role in the maintenance of amniotic fluid homeostasis, and its expression level can be decreased with the deepening of insulin resistance during pregnancy, which may explain the frequent abnormal amniotic fluid homeostasis in pregnant women with diabetes [46]. Pf4 may be a new marker for monitoring coagulation function in patients with GDM and may also be a risk factor for GDM during pregnancy [47,48]. Lanoix et al. [49] found that peptidylprolyl isomerase can be used as a protein reference marker for GDM placental research. Mcat may indirectly trigger GDM by participating in fatty acid biosynthesis during pregnancy [50]. 

[45]Dubé E, Ethier-Chiasson M, Lafond J. Modulation of cholesterol transport by insulin-treated gestational diabetes mellitus in human full-term placenta. Biol Reprod. 2013 Jan 17;88(1):16. doi: 10.1095/biolreprod.112.105619. PMID: 23221398.

[46]Bouvier D, Rouzaire M, Marceau G, Prat C, Pereira B, Lemarié R, et al. Aquaporins and Fetal Membranes From Diabetic Parturient Women: Expression Abnormalities and Regulation by Insulin. J Clin Endocrinol Metab. 2015 Oct;100(10):E1270-9. doi: 10.1210/jc.2015-2057. Epub 2015 Jul 24. PMID: 26207951.

[47]Li ZF, Shen WW. Detection of PF4, β-TG and LP-PLA2 in pregnant women with gestational diabetes mellitus and its clinical significance. Hebei Medical Journal. 2020;42(21):3249-3252. doi:10.3969/j.issn.1002-7386.2020.21.012

[48]Hagedorn M, Zilberberg L, Lozano RM, Cuevas P, Canron X, Redondo-Horcajo M, et al. A short peptide domain of platelet factor 4 blocks angiogenic key events induced by FGF-2. FASEB J. 2001 Mar;15(3):550-2. doi: 10.1096/fj.00-0285fje. Epub 2001 Jan 5. PMID: 11259363.

[49]Lanoix D, St-Pierre J, Lacasse AA, Viau M, Lafond J, Vaillancourt C. Stability of reference proteins in human placenta: general protein stains are the benchmark. Placenta. 2012 Mar;33(3):151-6. doi: 10.1016/j.placenta.2011.12.008. Epub 2012 Jan 13. PMID: 22244735.

[50]Zhang Y, Ye J, Fan J. Regulation of malonyl-CoA-acyl carrier protein transacylase network in umbilical cord blood affected by intrauterine hyperglycemia. Oncotarget. 2017 Sep 8;8(43):75254-75263. doi: 10.18632/oncotarget.20766. PMID: 29088862; PMCID: PMC5650417.

14. The reviewer’s comment: In the conclusion, it is stated that “The characteristics induced by the HFHS+MR modeling method were more in line with the pathological characteristics of GDM than those induced by the other modeling methods,” but in Lines 483-484, the authors state that, “only the rats in the HFHS+STZ group presented the obvious clinical manifestations of GDM with polydipsia, polyphagia, polyuria, and body weight gain and the typical pathogenesis of GDM with high IR levels and decreased insulin sensitivity.” For me these two statements do not align with each other. Perhaps the authors are making the point that the cause of the pathology in the HFHS+MR group is more similar to the etiology of human GDM, compared to the STZ-induced model? Please make this point clearer.

Our thoughts and corrections: Thanks to your comments, we have made this part of the conclusion clearer. The details are as follows (see the lines 581-591 in the Conclusion section on ):

Our findings indicated that the modified HFHS diet combined with a single intraperitoneal injection of 25 mg/kg STZ was the optimal method for constructing a rat model of nonspontaneous GDM. Compared with the other two models, the rat model of the HFHS+STZ group had the advantage of better representing the broad phenotype and pathology of human GDM, and the model featured high stability. Compared with the HFHS+STZ modeling method, the pathogeny simulation of the HFHS+MR group were more similar to the etiology of human GDM and also caused obvious pathological changes in rats, but the modeling rate was lower than that of other methods. In follow-up research, we will modify the method of movement restriction and physical activity measure, such as using telemetry devices, in order to construct a new rat model that better match the characteristics of human GDM. 

Looking forward to hearing from you.

Thank you and best regards.

Yours sincerely,

Li Ge

---

## [Editor Report · Decision Letter 1]

15 Aug 2022

Construction of the experimental rat model of gestational diabetes

PONE-D-21-38566R1

Dear Dr. Ge,

We’re pleased to inform you that your manuscript has been judged scientifically suitable for publication and will be formally accepted for publication once it meets all outstanding technical requirements.

Kind regards,

Michael Bader

Academic Editor

PLOS ONE
---

## [Editor Report · Acceptance letter]

2 Sep 2022

PONE-D-21-38566R1 

Construction of the experimental rat model of gestational diabetes 

Dear Dr. Ge:

I'm pleased to inform you that your manuscript has been deemed suitable for publication in PLOS ONE. Congratulations! Your manuscript is now with our production department. 

Kind regards, 

on behalf of

Prof. Michael Bader 

Academic Editor

PLOS ONE